# Epidemiological characteristics of imported acute infectious diseases in Guangzhou, China, 2005–2019

**Wen-Hui Liu[1,2]⊙, Chen Shi[2]⊙, Ying Lu[1]⊙, Lei Luo**📷**[1]\*, Chun-Quan Ou[2]\***

**1** Guangzhou Center for Disease Control and Prevention, Guangzhou, China, **2** State Key Laboratory of Organ Failure Research, Department of Biostatistics, Guangdong Provincial Key Laboratory of Tropical Disease Research, School of Public Health, Southern Medical University, Guangzhou, China

⊙ These authors contributed equally to this work.
\* llyeyq@163.com (LL); ouchunquan@hotmail.com (CO)

**Data Availability Statement:** The data that support the findings of this study are available from Guangzhou Center for Disease Control and Prevention but restrictions apply to the availability

## Abstract

### Background

The global spread of infectious diseases is currently a prominent threat to public health, with the accelerating pace of globalization and frequent international personnel intercourse. The present study examined the epidemiological characteristics of overseas imported cases of acute infectious diseases in Guangzhou, China.

### Methods

We retrospectively investigated the distribution of diseases, demographic characteristics, and temporal and spatial variations of imported cases of acute infectious diseases in Guangzhou based on the surveillance data of notifiable infectious diseases from 2005 to 2019, provided by Guangzhou center for Disease Control and Prevention. The Cochran-Armitage trend test was applied to examine the trend in the number of imported cases over time.

### Results

A total of 1,025 overseas imported cases of acute infectious diseases were identified during the study period. The top three diseases were dengue (67.12%), malaria (12.39%), and influenza A (H1N1) pdm09 (4.10%). Imported cases were predominantly males, with a sex ratio of 2.6: 1 and 75.22% of the cases were those aged 20–49 years. Businessmen, work-ers, students and unemployed persons accounted for a large proportion of the cases (68.49%) and many of the cases came from Southeast Asia (59.02%). The number of imported cases of acute infectious diseases increased during the study period and hit 318 in 2019. A clear seasonal pattern was observed in the number of imported cases with a peak period between June and November. Imported cases were reported in all of the 11 districts in Guangzhou and the central districts were more seriously affected compared with other districts.

of these data, and so the data are not publicly available. Permission can be requested by contacting Guangzhou Center for Disease Control and Prevention (gzcdccfk@gz.gov.cn).

**Funding:** This work was supported by the Key Project of Medicine Discipline of Guangzhou to WHL(2021-2023-11), the Basic Research Project of Key Laboratory of Guangzhou to WHL (202102100001) and Guangzhou Municipal Science and Technology Project, China to LL (202102080132). The funders had no role in study design, data collection and analysis, decision to publish, or preparation of the manuscript.

**Competing interests:** The authors have declared that no competing interests exist.

## Conclusions

The burden of dengue imported from overseas was substantial and increasing in Guangzhou, China, with the peak period from June to November. Dengue was the most common imported disease. Most imported cases were males aged 20–49 years and businessmen. Further efforts, such as strengthening surveillance of imported cases, paying close attention to the epidemics in hotspots, and improving the ability to detect the imported cases from overseas, are warranted to control infectious diseases especially in the center of the city with a higher population density highly affected by imported cases.

### Author summary

Guangzhou, a city located in the south of China, is heavily affected by imported infectious disease, with frequent trade, personnel movements and unique subtropical monsoon climate which favors transmissions of various infectious diseases. In this study, we examined the epidemiological characteristics of imported cases of acute infectious diseases in Guangzhou, 2005–2019. Our findings highlighted the potential risk of spread of infectious diseases triggered by imported cases. The top three imported acute infectious diseases were dengue, malaria and influenza A (H1N1)pdm09 and most of the imported cases came from Southeast Asian. The burden of dengue imported from overseas was substantial and increasing in Guangzhou, China, with the peak period from June to November. Most imported cases were males aged 20–49 years and businessmen. Further efforts, such as strengthening surveillance of imported cases, paying close attention to the epidemics in hotspots, and improving the ability to detect the imported cases from overseas, are warranted to control infectious diseases especially in the center of the city with a higher population density highly affected by imported cases.

## Background

Infectious diseases, which are known to have no boundaries, pose a serious threat to public health. Accelerating globalization and high population mobility facilitates the spread of infectious diseases worldwide [1]. In 2016, the number of international travelers reached 1.2 billion [2] and the total number of inbound and outbound travelers in China exceeded 250 million [3]. Such frequent movements have remarkably increased the risk of importing and exporting cases of infectious diseases for a country, posing a serious challenge to disease prevention and control.

China, the largest developing country in the world, is also at elevated risk of infectious diseases. A previous study showed that the number of imported infectious disease cases in China generally went up from around 1800 in 2005 to over 4000 in 2016 [4]. Some studies have reported the epidemiological characteristics of imported infectious diseases in China [4–9]. It was reported that most persons with imported cases were male [4–6], and main imported diseases in mainland China generally exhibited seasonality [4]. In China, the main imported diseases were mosquito-borne infectious diseases such as malaria and dengue and the burden of imported infectious diseases varied across provinces [4]. Distinct epidemiological characteristics of imported infectious diseases were reported in different study locations in China. For example, imported dengue fever during 2005–2016 were identified in 27 provinces across

China but not in Shanxi, Qinghai, Ningxia, or Tibet, possibly due to the disparities in the countries of origin of the imported cases and the climates of the locations [5,8]. Therefore, the investigation in different areas can help better understanding the epidemiologic characteristics of imported infectious diseases and provide reference for planning resource allocation in response to imported acute infectious diseases.

Guangzhou is located in the south of China, the hinterland of the Pearl River Delta, with a resident population of 14 million [10]. It is one of the China's four major economic centers. Frequent trade and personnel movements, coupled with the unique subtropical monsoon climate, make Guangzhou a high risk region with imported acute infectious diseases [10–12]. However, a full investigation into the imported infectious diseases in Guangzhou has not been conducted. The present study aimed to elucidate the epidemiological characteristics of imported cases of acute infectious diseases in Guangzhou from 2005 to 2019, helping for early identification and accurate prevention and control of these diseases.

## Methods

### Data collection

Data on cases of imported infectious diseases in Guangzhou were acquired from the China Information System for Disease Control and Prevention (CISDCP). CISDCP is the most important and basic macro surveillance system for infectious diseases in China established by Chinese Center for Disease Control and Prevention (China CDC) in 2004. The information of an infectious disease case included demographic characteristics (e.g. sex, age, occupation), the type of infectious disease, whether a case was detected by clinical diagnosis or laboratory confirmation, date of illness onset, country of origin, the district in which the case was reported. Due to incomplete records prior to 2005 and in order to exclude the impact of the COVID-19 on the social distance and the frequency of international communication in recent years, we choose 2005–2019 as the study period. All data were anonymized during the analysis. We obtained the population density of Guangzhou by district from the 6th National Census (http://tjj.gz.gov.cn/tjgb/glpcgb/content/post_2788677.html).

### Case definition

Acute infectious diseases are infectious diseases characterized by acute onset of symptoms, having a clear history of epidemiological exposure [13]. Infectious diseases were all diagnosed according to diagnosis criteria enacted by the Ministry of Health of the People's Republic of China. Imported cases refer to those who had a residence history in an epidemic area outside mainland China during the longest incubation period of the disease before the onset of symptoms [4]. That is imported cases include not only foreign visitors and migrant workers, but also citizens of China returning from overseas. Laboratory diagnosed or clinically diagnosed cases of imported acute infectious diseases were included, while suspected cases were excluded.

### Statistical analysis

The imported acute infectious diseases were classified as dengue, malaria, influenza A(H1N1) pdm09 and others. We described the distribution of the main imported infectious diseases, demographic characteristics of the imported cases, and temporal and spatial patterns in the number of imported cases. Specifically, we calculated the proportions of the four kinds of imported diseases across study years. Cochran-Armitage trend test was applied to test the trend in the number of imported cases over time [14]. Meanwhile, seasonal index was used to

describe the seasonal fluctuation in the number of imported cases. The index for a given month was calculated by the average case number of that month divided by the average monthly cases during the 15 years (2005–2019). Seasonal fluctuation is not obvious if the index in each month is closer to 1 [5]. We applied Getis-Ord General G to determine the spatial aggregation of imported cases of acute infectious diseases [15]. All analyses were performed using SAS 9.2, IBM SPSS Statistics 24.0 and R 4.1.1.

## Results

### Distribution of imported acute infectious diseases

A total of 1,025 overseas imported cases of 22 acute infectious diseases were reported in Guangzhou from 2005 to 2019. The top three diseases were dengue (688, 67.12%), malaria (127, 12.39%), and influenza A (H1N1)pdm09 (42, 4.10%) (Table 1). The proportion of dengue cases among the imported cases increased from 7.69% in 2005 to 92.52% in 2019, while the percentage of infectious diseases excluding dengue, malaria and influenza A (H1N1)pdm09 declined from 92.31% in 2005 to 2.18% in 2019 (Fig 1). During the study period, the proportion of malaria cases increased at first, peaking in 2010–2012, and then decreased. It was worth noting that the influenza A (H1N1)pdm09 cases accounted for the largest proportion of imported cases in 2009. Besides, there were some emerging infectious diseases imported from overseas. For example, two cases of Zika virus infection were imported from Venezuela and

**Table 1. The distribution of imported acute infectious diseases in Guangzhou, 2005–2019.**

| Disease | Number of cases | Percentage (%) |
| --- | --- | --- |
| Dengue | 688 | 67.12 |
| Malaria | 127 | 12.39 |
| Influenza A (H1N1)pdm09# | 42 | 4.10 |
| Hepatitis E | 36 | 3.51 |
| Influenza* | 28 | 2.73 |
| Chicken pox | 18 | 1.76 |
| Other infectious diarrhea | 15 | 1.46 |
| Chikungunya | 14 | 1.37 |
| Gonorrhea | 13 | 1.27 |
| Hepatitis A | 12 | 1.17 |
| Hand foot mouth disease | 9 | 0.88 |
| Mumps | 5 | 0.49 |
| Measles | 4 | 0.39 |
| Zika | 3 | 0.29 |
| Cholera | 2 | 0.20 |
| Acute hemorrhagic conjunctivitis | 2 | 0.20 |
| Bacillary dysentery | 2 | 0.20 |
| Epidemic hemorrhagic fever | 1 | 0.10 |
| Leptospirosis | 1 | 0.10 |
| Typhoid or paratyphoid fever | 1 | 0.10 |
| Scarlet fever | 1 | 0.10 |
| Scrub typhus | 1 | 0.10 |
| Total | 1,025 | 100.00 |

# The cases were imported during the 2009 pandemic period.

* The cases were imported in 2005, 2012 and 2017.

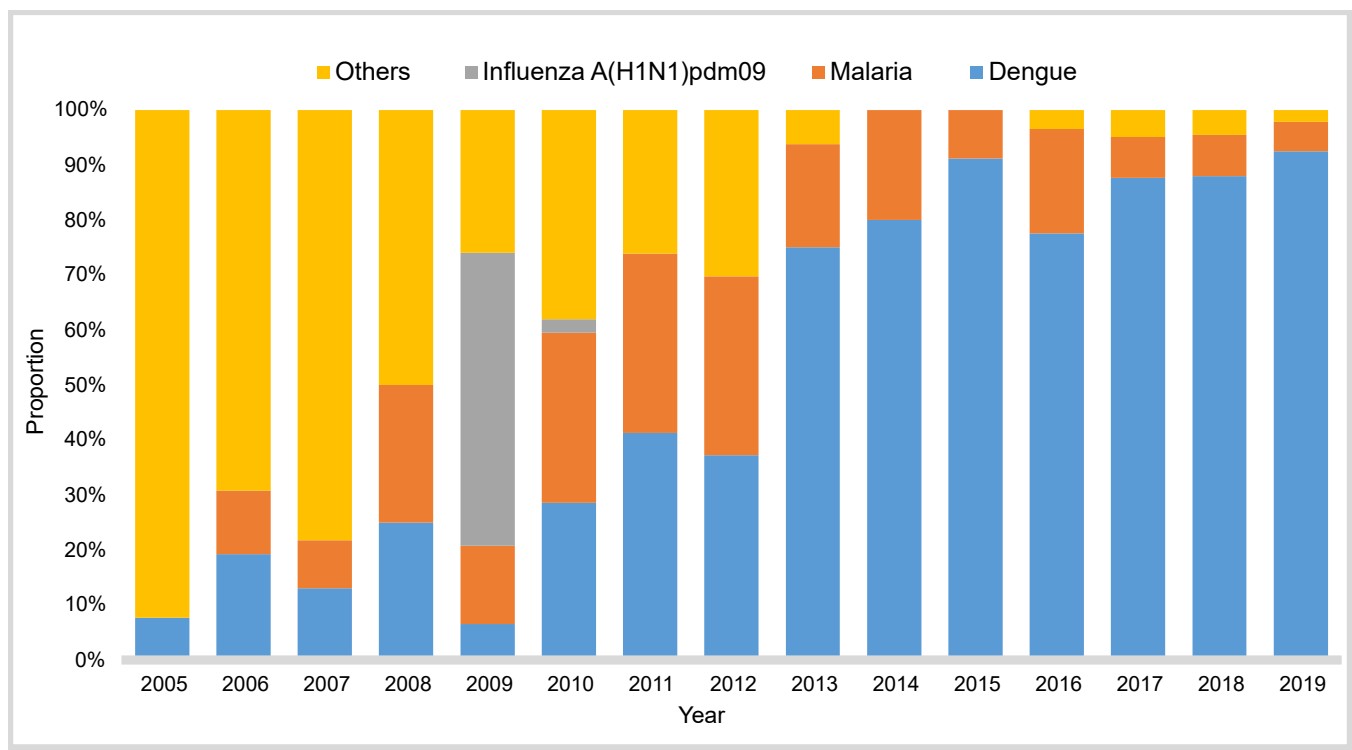

**Fig 1. Proportions of dengue, malaria, influenza A (H1N1)pdm09 and other imported cases of acute infectious diseases in Guangzhou, 2005–2019.**
Dengue was the most commonly imported cases with an increasing proportion over time, followed by malaria. The influenza A (H1N1)pdm09 cases accounted for the largest proportion of imported cases in 2009.

Suriname in 2016, and 17 cases of Chikungunya fever were imported from Africa and Southeast Asian during 2017–2019 (S2 Table).

## Demographic characteristics

Cases were predominantly males (72.10%, 740/1025) with a sex ratio of 2.6: 1. Adults aged 20–49 years accounted for 75.22% of the imported cases. Businessmen, workers, students and unemployed persons represented approximately 70% of the imported cases (Table 2).

Most of the imported cases came from Southeast Asian (59.02%, 605/1025), such as Cambodia and Thailand (Fig 2), which was followed by East Asia, South Asian and West Africa. Among the imported cases from Southeast Asian, 564 were dengue cases (93.22%). There were 219 dengue cases and 1 malaria case coming from Cambodia, while 115 dengue, 2 chikungunya fever, 2 influenza A (H1N1)pdm09 and 1 epidemic hemorrhagic fever cases were imported from Thailand.

## Temporal patterns

Overall, the number of cases imported from overseas in Guangzhou increased during 2005–2019 ($P<0.001$). A small peak was observed in 2009, with 31.32% of the imported cases occurring in this year. And the number of imported cases continued to rise after 2015 ($P<0.001$) (Fig 3). The peak period of imported acute infectious diseases in Guangzhou was from June to November. The seasonal fluctuation for dengue was more obvious than that for malaria (Fig 4).

**Table 2. Demographic characteristics of imported cases of acute infectious diseases in Guangzhou, China, 2005–2019.**

| Characteristics | Number of cases (%) | | | | |
|---|---|---|---|---|---|
| | Overall | Dengue | Malaria | Influenza A (H1N1) pdm09 | Others |
| Sex | | | | | |
| Male | 720 (72.20) | 474 (68.90) | 111 (87.40) | 28 (66.67) | 107 (72.30) |
| Female | 285 (27.80) | 214 (31.10) | 16 (12.60) | 14 (33.33) | 41 (27.70) |
| Age, years | | | | | |
| <5 | 21 (2.05) | 1 (0.15) | 0 (0.00) | 0 (0.00) | 20 (11.90) |
| 5–9 | 16 (1.56) | 2 (0.29) | 1 (0.79) | 1 (2.38) | 12 (7.14) |
| 10–14 | 18 (1.76) | 6 (0.87) | 1 (0.79) | 6 (14.29) | 5 (2.98) |
| 15–19 | 54 (5.27) | 23 (3.34) | 1 (0.79) | 3 (7.14) | 27 (16.07) |
| 20–29 | 262 (25.56) | 191 (27.76) | 32 (25.20) | 19 (45.24) | 20 (11.90) |
| 30–39 | 312 (30.44) | 239 (34.74) | 43 (33.86) | 7 (16.67) | 23 (13.69) |
| 40–49 | 197 (19.22) | 137 (19.91) | 32 (25.20) | 2 (4.76) | 26 (15.48) |
| ≥50 | 145 (14.15) | 89 (12.94) | 17 (13.39) | 4 (9.52) | 35 (20.83) |
| Occupation | | | | | |
| Businessman | 320 (31.22) | 235 (34.16) | 47 (37.01) | 8 (19.05) | 30 (17.86) |
| Unemployed person | 160 (15.61) | 131 (19.04) | 7 (5.51) | 5 (11.90) | 17 (10.12) |
| Worker | 129 (12.59) | 102 (14.83) | 17 (13.39) | 1 (2.38) | 9 (5.36) |
| Student | 93 (9.07) | 36 (5.23) | 7 (5.51) | 14 (33.33) | 36 (21.43) |
| Government official | 81 (7.90) | 60 (8.72) | 10 (7.87) | 4 (9.52) | 7 (4.17) |
| Traveler | 30 (2.93) | 27 (3.92) | 0 (0.00) | 0 (0.00) | 3 (1.79) |
| Child | 28 (2.73) | 1 (0.15) | 0 (0.00) | 1 (2.38) | 26 (15.48) |
| Farmer | 22 (2.15) | 16 (2.33) | 4 (3.15) | 1 (2.38) | 1 (0.60) |
| Others | 54 (5.27) | 10 (1.45) | 16 (12.60) | 8 (19.05) | 20 (11.90) |

## Spatial patterns

During the study period, all of the 11 districts in Guangzhou had imported cases of infectious diseases. Yuexiu, Panyu and Tianhe districts had more imported cases (720) than other eight districts (305). And imported cases were clustered in the center of the city ($P$ = 0.007) where the population density was higher than others (Fig 5).

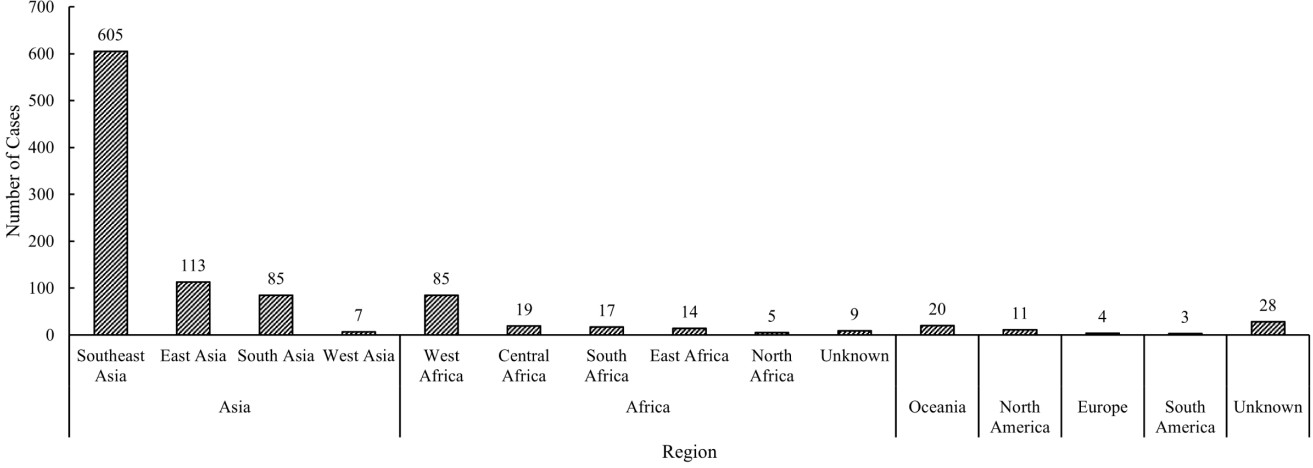

**Fig 2. The distribution of original countries of imported cases in Guangzhou, 2005–2019.** Most of the imported cases came from Southeast Asian.

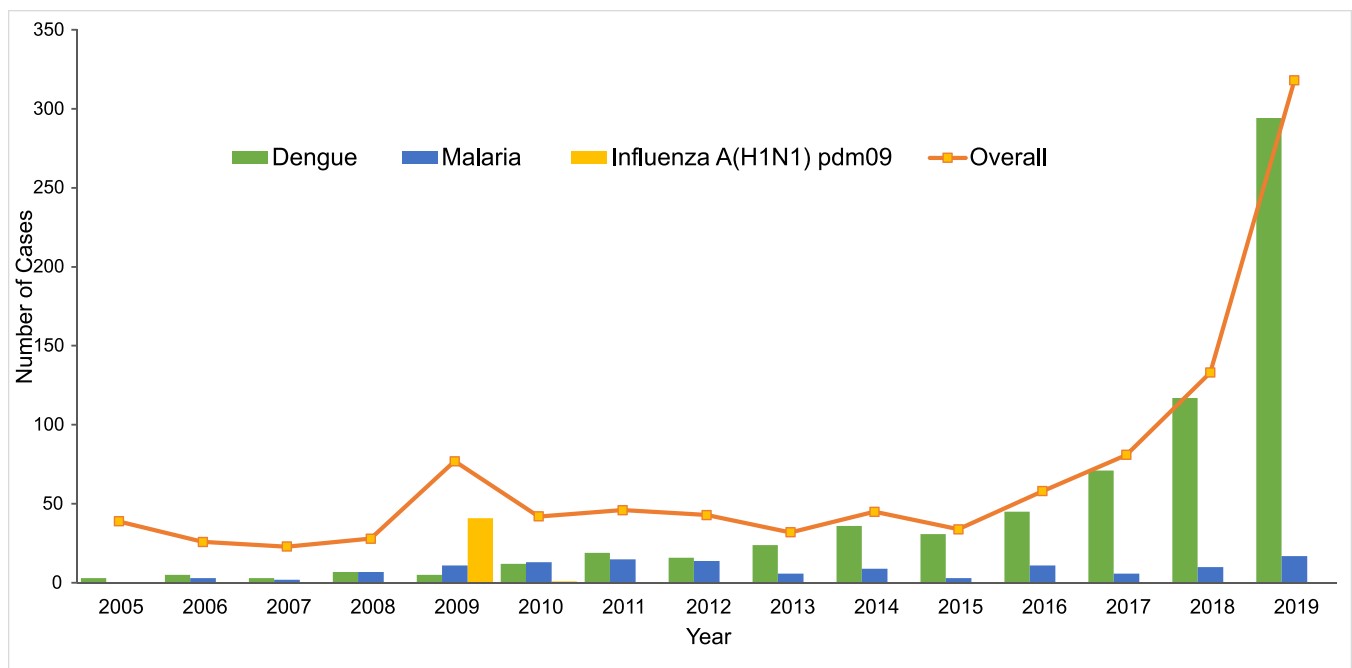

**Fig 3. The number of imported cases of infectious diseases in Guangzhou, 2005–2019.**

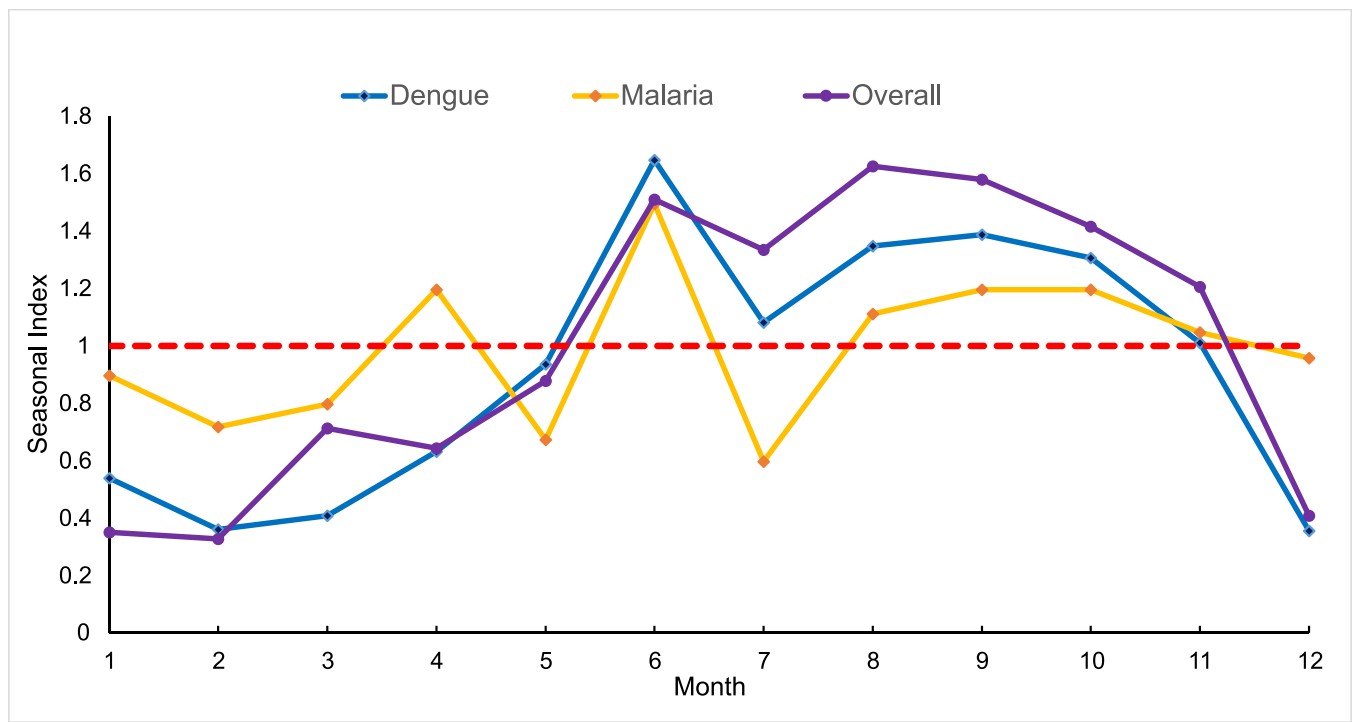

**Fig 4. Seasonal index of the average monthly number of imported cases of infectious diseases in Guangzhou, 2005–2019.**

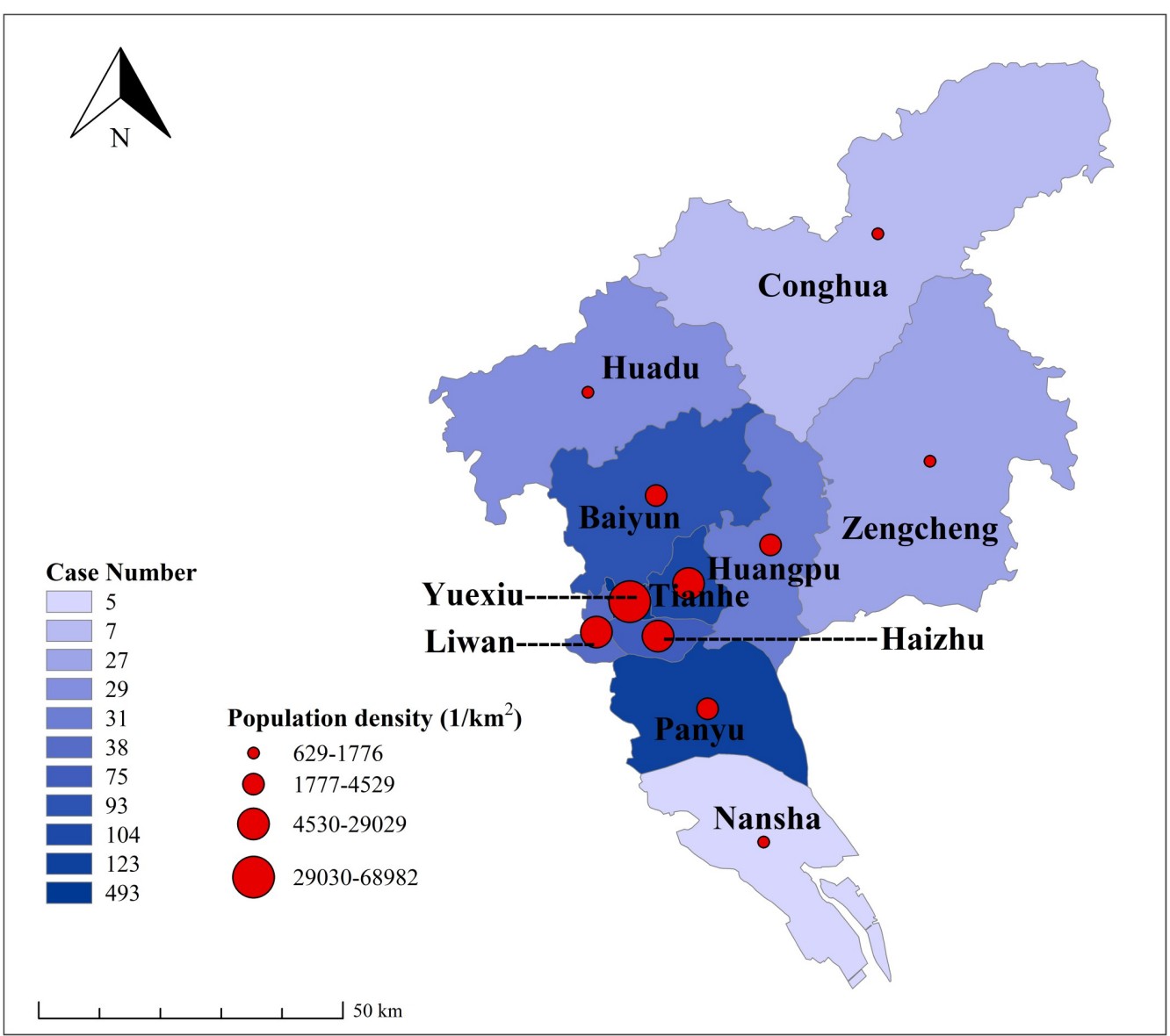

**Fig 5. The geographic distribution of imported cases and population density by district in Guangzhou, 2005–2019.** Basemap shapefile's map content from National Earth System Science Data Center, National Science & Technology Infrastructure of China (http://www.geodata.cn), approval number 272148515751668.

## Discussion

The present study examined the epidemiological characteristics of imported acute infectious diseases in Guangzhou during 2005–2019. We found that the number of cases imported from overseas in Guangzhou increased during the study period, mainly seen in males aged 20–49 years and in businessmen and immigration workers from Southeast, East, South Asia and West Africa. The findings were consistent with another study in China during 2013–2016 [16]. This study revealed that dengue was the most common disease and June to November were the epidemical period.

Guangzhou is adjacent to Southeast, East, and South Asia. According to the data from the National Tourism Administration, the top three countries in China's outbound tourism

destinations along "the Belt and Road" were Thailand, Singapore, and Malaysia [17]. Increasing number of outbound tourists could partially explain the large amounts of imported cases from these regions. In recent years, a certain number of African countries have participated in China's "Belt and Road" plan and many Chinese companies have set up in Africa for energy development, transportation and other infrastructure construction [18]. The huge export of labor and frequent trade created conditions for the import of acute infectious diseases. In addition, harsh field work, relatively poor living conditions and climatic characteristics in African countries further increased the risk of infection with acute infectious diseases such as dengue and malaria among Chinese people working there. Therefore, it is of great importance to strengthen the management of outbound tourists, businessmen and migrant workers especially those from hotspots. At the same time, the implementation of health education is also essential for people travelling abroad.

Since the outbreak of dengue was first reported in Hainan province in 1978, the number of dengue cases in China has been on the rise, with affected areas expanding. It has become China's most serious mosquito-borne infectious disease [12]. As the capital of Guangdong Province, the number of dengue cases in Guangzhou accounted for approximately 70% of cases in Guangdong Province and 50% of all cases in China from 1978 to 2011 [12,19]. Our finding revealed that dengue was one of the top imported acute infectious diseases in Guangzhou during 2005–2019, which was consistent with the reports of imported infectious diseases in Yunnan Province and Zhejiang Province [5,8]. Dengue in Guangzhou is considered as an epidemic mode of local spread caused by imported epidemics [20,21]. The rise in imported cases is expected to increase the overall intensity of the dengue epidemic in Guangzhou. There was a high peak of reported dengue cases in 2014 in Guangzhou. Although only 36 out of the total 37359 cases in 2014 were imported cases from outside China (S1 Table), many cases were domestically transmitted from the adjacent cities like Foshan, Zhongshan where the first dengue case in 2014 was also imported by oversea travel [10]. Besides, extraordinary high precipitation in May and August, 2014 increased vector abundance, which means the environment can support more mosquitoes that transmit viruses [22]. The government paid more attention to early detection of imported cases, early mosquito control and the quarantine of suspicious cases after the unprecedented outbreak in 2014. Therefore, the outbreak size after 2014 became relatively small even though the imported cases increased [22].

The number of imported malaria cases remained at a relatively low level in Guangzhou with the number of annual imported cases ranging from 0 to 17 during the study years. There has been no indigenous case of malaria in China since August 2016 [23]. Coupled with the implementation of screening for malaria infections at borders, airports and ports, the risk of local transmission of imported malaria cases has been reduced significantly. It is worth noting that Guangzhou has imported some emerging infectious diseases from overseas in recent years. For example, 2 cases of Zika were imported from Venezuela and Suriname in 2016, and 17 cases of Chikungunya fever were imported from Africa and Southeast Asian countries in 2017–2019 (S2 Table). This has brought new challenges to the prevention and control of infectious diseases.

From the perspective of temporal distribution in the number of imported cases, there was a small peak in 2009, while a sharp increase was observed between 2015 and 2019. On April 25, 2009, the World Health Organization (WHO) first announced the 2009 influenza pandemic as a public health emergency of international concern after the revision of the International Health Regulations (2005 edition) [24]. Affected by the pandemic, the number of imported influenza A(H1N1)pdm09 accounted for 53.25% of all imported infectious diseases in Guangzhou in 2009 (Fig 1). After 2015, the total number of imported cases in Guangzhou increased and around 87% of them were dengue (Fig 1), which was in line with the dengue epidemic in Southeast Asia in recent years [25]. In addition, the peak period of acute imported infectious

diseases in Guangzhou was June-November, it is recommended that the customs and other port departments should strengthen the health quarantine, laboratory inspection and notification of information of fever cases during the peak period of the dengue epidemic. In addition, emergency response and case tracking are required for early identification of the cases and accurate prevention and control of dengue.

In terms of geographical distribution, the regions with the largest number of imported cases in Guangzhou were Yuexiu, Panyu, and Tianhe districts. Yuexiu and Tianhe districts are economically developed, with more businessmen and migrant workers, while Panyu district has a close connection with foreign countries and it has a university town. These areas are in or near the center of city, and higher population density could further exacerbate the risk of local transmission of dengue from imported cases in these areas [26,27]. The Guangzhou municipal government is suggested to reinforce close monitoring of imported cases, increase human and material investment in key areas, and establish a highly sensitive emergency response mechanism in order to control the spread of imported acute infectious diseases [28].

This study illustrated the severe situation of imported infectious cases in Guangzhou. The screening and detection of imported infectious diseases should be strengthened and improved. Specifically, at first, it is essential to strengthen training of medical staff to improve the awareness and ability to identify and diagnose imported cases. Second, the customs authority should enhance the health monitoring and screening of people entering from endemic countries. If they have fever and other relevant symptoms, in addition to tests for SARS-CoV-2 infection, it is recommended to screen arthropod-borne infectious diseases such as dengue fever. Third, during the epidemic season of dengue fever in Guangzhou, rapid antigen testing can also be considered on inbound persons with a history of residence in the areas with a high prevalence of dengue fever, and the information should be promptly updated to the community for strengthening surveillance in their districts. Forth, it is also important to highlight the publicity and health education on common imported infectious diseases among entry-exit persons.

Several limitations of this study should be mentioned. First, the surveillance of imported cases in Guangzhou mainly focus on plague, cholera, yellow fever, dengue, malaria and other key diseases. Not all of the infectious diseases are included in the surveillance of imported cases. Monitoring reports might underestimate the burden of imported infectious diseases overseas. Second, some asymptomatic infections may not be detected by the current surveillance. Third, due to the different incubation periods of individuals, especially for those who travelled abroad for a short period of time, the data might be biased toward the identification of an imported case rather than a local case.

## Conclusions

There was a substantial and increasing burden of imported acute infectious cases in Guangzhou, China. Dengue was the most common disease and June to November were the epidemical period. Most imported cases were adults and businessmen. Further efforts, such as strengthening surveillance of imported cases, paying close attention to the epidemics in hotspots, and improving the ability to detect the imported cases from overseas, are warranted to control infectious diseases especially in the center of the city with a higher population density highly affected by imported cases.

## Supporting information

**S1 Table. The number of total reported cases and imported cases of dengue in Guangzhou, 2005–2019.**
(XLSX)

**S2 Table. The distribution of original countries of all imported cases in Guangzhou, 2005–2019.**
(XLSX)

## Acknowledgments

We acknowledged the hard work of district-level disease control and prevention institutions and community health service centers.

## Author Contributions

**Conceptualization:** Wen-Hui Liu, Ying Lu, Lei Luo.

**Data curation:** Chen Shi.

**Formal analysis:** Wen-Hui Liu, Chen Shi.

**Funding acquisition:** Lei Luo.

**Methodology:** Chun-Quan Ou.

**Software:** Wen-Hui Liu.

**Supervision:** Lei Luo, Chun-Quan Ou.

**Writing – original draft:** Chen Shi, Ying Lu.

**Writing – review & editing:** Chen Shi, Ying Lu, Lei Luo, Chun-Quan Ou.

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
