## [Decision Letter · Decision Letter 0]

8 Aug 2022

Dear Dr Luo,

Thank you very much for submitting your manuscript "Epidemiological characteristics of imported acute infectious diseases in Guangzhou, China, 2005-2019" for consideration at PLOS Neglected Tropical Diseases. As with all papers reviewed by the journal, your manuscript was reviewed by members of the editorial board and by several independent reviewers. In light of the reviews (below this email), we would like to invite the resubmission of a significantly-revised version that takes into account the reviewers' comments. 

Dear Authors, thank you for submitting in Plos NTD. Your manuscript has been assessed by relevant experts from the field. They found the manuscript interesting but raised some concerns in methodology and interpretation of results. It is requested to please consider the comments of reviewers.

We cannot make any decision about publication until we have seen the revised manuscript and your response to the reviewers' comments. Your revised manuscript is also likely to be sent to reviewers for further evaluation.

Sincerely,

Tauqeer Hussain Mallhi, Ph.D

Academic Editor

Waleed Al-Salem

Section Editor

Dear Authors, thank you for submitting in Plos NTD. Your manuscript has been assessed by relevant experts from the field. They found the manuscript interesting but raised some concerns in methodology and interpretation of results. It is requested to please consider the comments of reviewers.

Reviewer's Responses to Questions

**Key Review Criteria Required for Acceptance?**

**Methods**

-Are the objectives of the study clearly articulated with a clear testable hypothesis stated?

-Is the study design appropriate to address the stated objectives?

-Is the population clearly described and appropriate for the hypothesis being tested?

-Is the sample size sufficient to ensure adequate power to address the hypothesis being tested?

-Were correct statistical analysis used to support conclusions?

-Are there concerns about ethical or regulatory requirements being met?

Reviewer #1: The importance and objectives of the study were nicely illustrated in the background section. As this is a retrospective study I think the study design was appropriate as expected in this type of epidemiological study. Sample size is alright considering the case definition and to draw the conclusions they did. 

The statistical tests that has been done looks ok. Still, I think you should have mentioned the definition of "Seasonal Index" and how you calculated this in the method section.

Reviewer #2: The methodology in this study is not very clear and requires major revisions, which are suggested below.

Reviewer #3: The objectives are clear and appropriately addressed. Minor comments:

1. Please add 'Imported cases' as key words. And remove the word imported from 'imported acute infectious diseases'

2. Add the definition of imported case in the case definition section (with reference). Add the references for 'Acute infectious diseases'

**Results**

-Does the analysis presented match the analysis plan?

-Are the results clearly and completely presented?

-Are the figures (Tables, Images) of sufficient quality for clarity?

Reviewer #1: The analysis presented looks good according to the analysis plan and are clearly stated in the result section. 

I have few concerns for this section:

1) The figures provided are blur and not clear. Please provide a better version of each plot.

2) Another analysis would be wonderful to include, which is a correlation analysis between major diseases and source (origin) of the disease. This analysis would be helpful for the government during management of situation or control the spread of the disease. 

3) In the supporting information file you provided a table containing "The number of reported cases of dengue in Guangzhou, 2005-2019". According to that, there was an exceptionally high peak of reported dengue cases in 2014 (i.e. 37359). But you did not explain whether this issue inside the paper. You should explain if this was caused by imported cases and how. And if this is completely domestic epidemic your statement in the discussion section (line 180-181) is not correct. Please explain.

4) In the line 187 of the discussion section you mentioned "study years (0~17)". What does it mean. If this is a typo, please correct it.

Reviewer #2: Major revisions are required when it comes to the results section of the paper. The figures do not have any legends associated with them. The legends are mentioned in the results section instead.

Reviewer #3: Good analysis. minor comment.

In table 2, add an additional column for other case.

**Conclusions**

-Are the conclusions supported by the data presented?

-Are the limitations of analysis clearly described?

-Do the authors discuss how these data can be helpful to advance our understanding of the topic under study?

-Is public health relevance addressed?

Reviewer #1: Conclusions are mostly supported by the data presented. Authors also mentioned the major limitations of the study. They also discussed the public health importance and relevance.

Reviewer #2: The conclusion section does not have references and re-writing of the section is highly suggested.

Reviewer #3: limitations of analysis are clearly described. 

Please add some points regarding strengthening of the detection procedure for imported cases.

**Editorial and Data Presentation Modifications?**

Reviewer #1: Minor revision

Reviewer #2: Line 20 :- "...Southeast Asian..." should be re-written as "...Southeast Asia..."

Line 47 :- "Infectious diseases, which are known to have no boundaries..."

Line 49 :- Please replace "globally" with "worldwide".

Line 49 :- Please remove the word " worldwide".

Line 50 :- What do you mean by Chinese entry and exit? Please rewrite that.

Line 50-51 :- Please remove " mass population".

Line 54 :- "Increasing rapid and mass population movement"--very repetitive. Please rewrite that. 

Line 55 :- Please re-write the entire sentence. 

Line 55 :- What is the "Belt and Road" initiative?

Line 56 :- Please remove the word "importation".

Line 57 :- Can you elaborate with numbers, that the number of cases went up from 2005-2016?

Line 58-59 :- "The assessment.... measures"--please remove the line. 

Line 59-60 :- Please explain the epidemiological characteristics of imported infectious diseases in China, in detail. 

Line 62-64 :- Please elaborate. 

Line 66 :- Please add a reference. 

Line 68 :- Please replace "prone-prone"

Line 69 :- Please rewrite "...diseases in Guangzhou has not been investigated..."

Line 71 :- Why were these specific years 2005-2019 chosen for the study? 

Line 72 :- The words "acute imported infectious diseases" ---very repetitive. 

Line 86-87 :- Very repetitive. Please re-write. 

Line 87 :- What is "unified national diagnostic criteria"?

Line 92 :- The words "acute imported infectious diseases" ---very repetitive. 

Line 101-102 :- Please rewrite. 

Line 156-157 :- The words "acute imported infectious diseases" ---very repetitive. 

Line 157 :- "...Guangzhou were mainly seen in males, aged 20-49 years..."

Line 160-161 :- Please elaborate this sentence. 

Line 163 :- Please add a reference. 

Line 198 :- Please add a reference.

Reviewer #3: (No Response)

**Summary and General Comments**

Reviewer #1: I think the major weakness of the study is that it is based on the symptomatic cases only. However, it was a well-organized manuscript with clear objectives, and I think, the authors have are mostly successful to achieve their goal.

Reviewer #2: (No Response)

Reviewer #3: The manuscript is nicely presented. Figures are appropriate.

PLOS authors have the option to publish the peer review history of their article (what does this mean?). If published, this will include your full peer review and any attached files.

Reviewer #1: No

Reviewer #2: No

Reviewer #3: No
---

## [Decision Letter · Decision Letter 1]

9 Nov 2022

Dear Dr Luo,

We are pleased to inform you that your manuscript 'Epidemiological characteristics of imported acute infectious diseases in Guangzhou, China, 2005-2019' has been provisionally accepted for publication in PLOS Neglected Tropical Diseases.

Best regards,

Hailey Schultz

Staff

Waleed Al-Salem

Section Editor

Reviewer's Responses to Questions

**Key Review Criteria Required for Acceptance?**

**Methods**

-Are the objectives of the study clearly articulated with a clear testable hypothesis stated?

-Is the study design appropriate to address the stated objectives?

-Is the population clearly described and appropriate for the hypothesis being tested?

-Is the sample size sufficient to ensure adequate power to address the hypothesis being tested?

-Were correct statistical analysis used to support conclusions?

-Are there concerns about ethical or regulatory requirements being met?

Reviewer #1: After the revision the manuscript is quite improved and good for publication

Reviewer #2: (No Response)

Reviewer #3: (No Response)

**Results**

-Does the analysis presented match the analysis plan?

-Are the results clearly and completely presented?

-Are the figures (Tables, Images) of sufficient quality for clarity?

Reviewer #1: Yes, result section meets all the required criteria

Reviewer #2: (No Response)

Reviewer #3: (No Response)

**Conclusions**

-Are the conclusions supported by the data presented?

-Are the limitations of analysis clearly described?

-Do the authors discuss how these data can be helpful to advance our understanding of the topic under study?

-Is public health relevance addressed?

Reviewer #1: Conclusions are satisfactory and meets the goal of the study

Reviewer #2: (No Response)

Reviewer #3: (No Response)

**Editorial and Data Presentation Modifications?**

Reviewer #1: (No Response)

Reviewer #2: (No Response)

Reviewer #3: (No Response)

**Summary and General Comments**

Reviewer #1: (No Response)

Reviewer #2: (No Response)

Reviewer #3: The authors addressed all the comments properly.

PLOS authors have the option to publish the peer review history of their article (what does this mean?). If published, this will include your full peer review and any attached files.

Reviewer #1: **Yes: **Hasan Al Banna

Reviewer #2: No

Reviewer #3: No

---

## [Editor Report · Acceptance letter]

18 Nov 2022

Dear Dr Luo,

We are delighted to inform you that your manuscript, "Epidemiological characteristics of imported acute infectious diseases in Guangzhou, China, 2005-2019," has been formally accepted for publication in PLOS Neglected Tropical Diseases.

Best regards,

Shaden Kamhawi

co-Editor-in-Chief

Paul Brindley

co-Editor-in-Chief
